# Novel *Ralstonia* species from human infections: improved matrix-assisted laser desorption/ionization time-of-flight mass spectrometry-based identification and analysis of antimicrobial resistance patterns

Stephanie Steyaert,[1] Charlotte Peeters,[1,2] Anneleen D. Wieme,[1,3] Astrid Muyldermans,[2,4] Kristof Vandoorslaer,[4] Theodore Spilker,[5] Ingrid Wybo,[2,4] Denis Piérard,[2,4] John J. LiPuma,[5] Peter Vandamme[1,2]

**ABSTRACT** A collection of 161 *Ralstonia* isolates, including 90 isolates from persons with cystic fibrosis, 27 isolates from other human clinical samples, 8 isolates from the hospital environment, 7 isolates from industrial samples, and 19 environmental isolates, was subjected to matrix-assisted laser desorption/ionization time-of-flight mass spectrometry (MALDI-TOF MS) identification and yielded confident species level identification scores for only 62 (39%) of the isolates, including four that proved misidentified subsequently. Whole-genome sequence analysis of 32 representative isolates for which no confident MALDI-TOF MS species level identification was obtained revealed the presence of seven novel *Ralstonia* species, including three and four that were isolated from cystic fibrosis or other human clinical samples, respectively, and provided the basis for updating an *in-house* MALDI-TOF MS database. A reanalysis of all mass spectra with the updated MALDI-TOF MS database increased the percentage of isolates with confident species level identification up to 77%. The antimicrobial susceptibility of 30 isolates mainly representing novel human clinical and environmental *Ralstonia* species was tested toward 17 antimicrobial agents and demonstrated that the novel *Ralstonia* species were generally multi-resistant, yet susceptible to trimethoprim/sulfamethoxazole, ciprofloxacin, and tigecycline. An analysis of genomic antimicrobial resistance genes in 32 novel and publicly available genome sequences revealed broadly distributed beta-lactam resistance determinants.

**IMPORTANCE** The present study demonstrated that a commercial matrix-assisted laser desorption/ionization time-of-flight mass spectrometry identification database can be tailored to improve the identification of *Ralstonia* species. It also revealed the presence of seven novel *Ralstonia* species, including three and four that were isolated from cystic fibrosis or other human clinical samples, respectively. An analysis of minimum inhibitory concentration values demonstrated that the novel *Ralstonia* species were generally multi-resistant but susceptible to trimethoprim/sulfamethoxazole, ciprofloxacin, and tigecycline.

**KEYWORDS** *Ralstonia*, MALDI-TOF MS, antimicrobial resistance, cystic fibrosis

*R*alstonia bacteria have emerged as opportunistic human pathogens that cause nosocomial infections in immunocompromised patients, especially in persons with cystic fibrosis (CF) (1–5). This genus comprises three species of human clinical interest, i.e., *Ralstonia insidiosa* (6), *Ralstonia mannitolilytica* (7), and *Ralstonia pickettii* (8, 9); four plant pathogenic species, i.e., *Ralstonia nicotianae*, *Ralstonia pseudosolanacearum*,

Address correspondence to Charlotte Peeters, charlotte.peeters@ugent.be.

Stephanie Steyaert and Charlotte Peeters contributed equally to this article. Author order was determined in order of increasing seniority.

The authors declare no conflict of interest.

*Ralstonia solanacearum*, and *Ralstonia syzygii* (9–14); and four environmental species, i.e., *Ralstonia chuxiongensis*, *Ralstonia mojiangensis*, *Ralstonia soli*, and *Ralstonia wenshanensis* (15, 16).

The reported prevalence of *Ralstonia* bacteria in persons with cystic fibrosis is generally less than 5%, but is increasing with *R. mannitolilytica* as the most frequently identified species, followed by *R. pickettii* and *R. insidiosa* (5, 17–19). Improved identification through the introduction of matrix-assisted laser desorption/ionization time-of-flight mass spectrometry (MALDI-TOF MS) as a key diagnostic tool in clinical laboratories likely contributed to this increase in prevalence (20). Commercial MALDI-TOF MS databases failed to identify many *Ralstonia* isolates at the species level and a recent whole-genome sequence-based analysis revealed that both clinical and environmental *Ralstonia* isolates were not accurately identified at the species level through MALDI-TOF MS (21–23). *Ralstonia* bacteria are commonly multi-resistant to antimicrobial therapy, with trimethoprim/sulfamethoxazole and fluoroquinolone antibiotics as the best treatment options (17, 23). Yet, there are important species-specific differences. In particular, *R. mannitolilytica* appeared to be the most resistant species, but resistance mechanisms have not been fully uncovered (17, 23).

The present study aimed to improve the diagnosis of human *Ralstonia* infections using MALDI-TOF MS and characterize the antimicrobial resistance patterns of novel *Ralstonia* species. A collection of 161 *Ralstonia* isolates, including 90 isolates from persons with CF, 27 isolates from other human clinical samples, 8 isolates from a hospital environment, 7 isolates from industrial samples, and 19 environmental isolates, was subjected to MALDI-TOF MS dereplication and identification. Representative isolates for which no unequivocal identification was obtained were selected for whole-genome sequence analysis. Genomic analyses revealed the presence of seven novel *Ralstonia* species, including three and four that were isolated from CF or other human clinical samples, respectively, which formed the basis for improving our *in-house* MALDI-TOF MS database. Finally, genomic antimicrobial resistance determinants and minimum inhibitory concentrations (MICs) of 17 antimicrobial agents toward a selection of human clinical and environmental *Ralstonia* isolates were determined.

## MATERIALS AND METHODS

### Bacterial isolates and growth conditions

All isolates analyzed are listed in Table S1. Historical and taxonomic reference strains were obtained from the Belgian Coordinated Collections of Microorganisms/Laboratory of Microbiology (BCCM/LMG) bacteria collection. Other isolates were from the authors' (J.L.P., D.P., and P.V.) research collections; these isolates (one per CF patient) originated from different patients and most likely represented different strains and were identified as described earlier (2, 6, 24). Isolates were grown on Tryptone Soya Agar (TSA) (Thermo Scientific PO5012A) and incubated aerobically for 48 h at 28°C, except for LMG 18093, which was incubated at 37°C. Cultures were preserved in MicroBank vials at −80°C.

### MALDI-TOF MS analysis

MALDI-TOF MS analysis was performed using the Bruker Microflex LT/SH Smart platform (Bruker Daltonik, Bremen, Germany). Isolates were grown as described above and subcultivated twice before harvesting the cell material. MALDI-TOF MS sample preparation, protein extraction, and data acquisition were performed as described previously (25).

For identification, mass spectra were compared to those in the commercial Bruker (IVD v12.0, MSP-11758) and *in-house* database (RUO, MSP-5331) using MBT Compass Explorer v4.1.80 software (Bruker Daltonics, Bremen, Germany). Identification of the isolates at genus and species level was based on the Bruker identification scores as specified in the MBT Compass Explorer software: Bruker scores between 2.300 and 3.000

were accepted for species-level identification and Bruker scores between 1.700 and 2.299 were accepted for genus-level identification. Isolates with Bruker scores below 1.699 were considered unidentified.

Mass spectra were converted to text format using FlexAnalysis v3.4 (Bruker Daltonics, Bremen, Germany). The text files were imported in BioNumerics v7.6.3. (Applied Math's, Sint-Martens-Latem, Belgium) and converted to fingerprints. The similarity between the mass spectra was expressed using the curve-based Pearson's product-moment correlation coefficient, and a dendrogram was built using the unweighted pair group method with the arithmetic mean clustering algorithm. Delineation of mass spectra in clusters based on global similarity (26, 27) was performed by visually inspecting the mass spectra in the dendrogram while taking into account the identification scores. Mass spectra were additionally dereplicated based on unique spectral features into mass spectrometry-defined independent isolates using the SPeDE algorithm with default settings (25). Final MALDI-TOF MS identification was based on Bruker identification scores in combination with clustering results.

## MALDI-TOF MS reference database construction

New main spectra (MSPs) were created using MBT Compass Explorer v4.1.80 software according to the manufacturer's instructions. To this end, isolates were subcultivated to the third generation, and protein extracts were prepared as described previously (25). The raw mass spectra of each isolate were analyzed in FlexAnalysis v3.4. Raw mass spectra with peak maxima falling outside the range of 500 ppm and aberrant spectra were removed. A minimum of 24 raw mass spectra were required for each isolate to create MSPs.

## DNA extraction

Cells were suspended in 300 µL of 4 M UltraPure guanidine isothiocyanate solution (Invitrogen 15577018). Genomic DNA was extracted using the Maxwell RSC Cultured Cells DNA kit (AS1620, Promega, USA) and the Maxwell RSC instrument (AS4500, Promega, USA) according to the manufacturer's instructions, except for the final step, in which the TE buffer was replaced by 10 mM Tris-HCl pH 8.5 elution buffer. DNA extracts were treated with RNase (2 mg/mL, 5 µL per 100 µL extract) and incubated at 37°C for 1 h. DNA quality was checked on a 1% agarose gel. DNA quantification was performed using the QuantiFluor ONE dsDNA system and the Quantus fluorometer (Promega, USA).

## Whole-genome sequencing

Whole-genome sequences were determined using the Illumina HiSeq 4000 platform at the Oxford Genomics Centre (Oxford, UK) or the NextSeq 2000 platform at MiGS Center (Pittsburgh, USA). Quality of raw data (PE150) was assessed with FastQC version 0.11.9 (28). Prior to assembly, reads were trimmed (Phred score >Q30) and filtered (length >50 bp) with fastp 0.23.2 (29) with correction option enabled. Assembly was performed using Shovill v1.1.0 (30) with SPAdes genome assembler 3.15.4 (31) at its core and read error correction disabled. Contigs shorter than 500 bp were removed from the final assembly. The quality of the final assembly and its summary statistics, such as the number of contigs, N50, L50, and the percentage G + C content were verified with QUAST (32). Finally, the assembly was checked for completeness and contamination using CheckM v1.1.2 (33). Annotation was performed using Prokka v1.14.5 (34). The annotated genome assemblies were submitted to the European Nucleotide Archive (ENA) and are publicly available under the accession numbers PRJEB43925 and PRJEB63170. Whole-genome sequences of the type strains of established *Ralstonia* species and *Cupriavidus necator* N-1[T] were downloaded from the National Center for Biotechnology Information database.

## Genomic taxonomy

Genomes were submitted to the Type (Strain) Genome Server (TYGS) (35, 36) to identify the nearest phylogenomic neighbors and calculate the degree of relatedness toward the nearest-neighbor species. Digital DNA–DNA hybridization (dDDH) values and confidence intervals were calculated using the recommended settings of GGDC 2.1 (37). The average nucleotide identity (ANI) values were calculated using FastANI (38). Genomes were also classified within the Genome Taxonomy Database using GTDB-tk v2.3.0 release 214 (39, 40). Whole-genome phylogeny was assessed based on 107 single-copy core genes found in the majority of bacteria (41) using bcgTree (42). Visualization and annotation of the phylogenetic tree were performed using iTOL (43).

## Antimicrobial resistance and virulence genes

All genomes were searched for antimicrobial resistance and virulence determinants using ABRicate (44) with the Comprehensive Antibiotic Resistance Database (CARD) (45) and Virulence Factor Database (VFDB) (46) reference databases. The Beta-Lactamase Database (BLDB) was used for classification of beta-lactamases into subfamilies (47).

## MIC analysis

MIC values for amikacin, tobramycin, amoxicillin, amoxicillin–clavulanic acid, meropenem, aztreonam, piperacillin, piperacillin–tazobactam, ceftolozane–tazobactam, temocillin, cefepime, ceftazidime, ceftazidime–avibactam, ciprofloxacin, and colistin were determined by microdilution in microtiter plates (Begnuz3, Thermo Scientific Sensititre). Plates were inoculated using a 0.5 McFarland suspension of cells from 24 h-old colonies grown on TSA. Ten µL of cell suspension was diluted in 11 mL of Mueller Hinton broth with TES buffer (Thermo Fisher Scientific), and 50 µL was dispensed in the microtiter plates using a Sensititre Automated Inoculation Delivery System (Thermo Fisher Scientific). MIC values were recorded using a Sensititre Vizion System (Thermo Scientific) after 24 h at 37°C as the lowest antibiotic concentration where growth was absent. If no growth was observed in the positive growth control well, the results were considered invalid.

MIC determination using E-tests (Liofilchem) was performed for tigecycline and trimethoprim/sulfamethoxazole on Mueller Hinton agar (Thermo Scientific R04052). Mueller Hinton agar plates were inoculated with a 0.5 McFarland cell suspension using a sterile cotton swab to obtain confluent growth. MIC values were recorded after 24 h at 37°C at 80% growth inhibition.

Interpretation of the MIC values was based on EUCAST clinical breakpoints (https:// www.eucast.org/clinical_breakpoints/, v13.1). As no species-specific breakpoints were available for *Ralstonia*, the *in vitro* susceptibility was interpreted using EUCAST pharmacokinetic/pharmacodynamic (PK/PD) breakpoints. Breakpoints for colistin were based on those from EUCAST *Pseudomonas* spp., and breakpoints for trimethoprim/sulfamethoxazole were based on those from EUCAST *Acinetobacter* spp..

## Phenotypic and biochemical characterization

Cell and colony morphology were assessed after 24 and 48 h of incubation at 28°C on TSA. Growth and biochemical characteristics were assessed using conventional procedures (48–50) with adapted concentrations of skim milk (2.8%) and gelatin (4%).

## Data visualization

Results from TYGS, FastANI, ABRicate, MIC, and phenotypic analyses were imported in R 4.1.3 and analyzed using tidyverse, janitor, igraph, Matrix, ape, ggnewscale, scales, and ggtext packages using Rstudio.

## RESULTS

### MALDI-TOF MS analyses

Upon analysis of all 161 *Ralstonia* isolates with the commercial Bruker database (IVD v12.0, MSP-11758, including nine *R. insidiosa* isolates, five *R. mannitolilytica* isolates, nine *R. pickettii* isolates, one *R. syzygii*, and two *Ralstonia* sp. isolates), the identification scores allowed the identification of only 62 isolates (39%) at the species level. All other isolates were identified as *Ralstonia* sp. (Table S2). Mass spectrum dereplication using the global pattern similarity and unique spectral features revealed 39 and 47 clusters of isolates, respectively (Table S2). Both dereplication data sets were used to select 32 representative isolates for whole-genome sequence-based identification (see below) (Table S2), which revealed the presence of isolates belonging to seven novel *Ralstonia* species and to two recently described environmental *Ralstonia* species, i.e., *R. chuxiongensis* and *R. wenshanensis*.

The *in-house* MALDI-TOF MS database was subsequently updated to include MSPs of (i) 19 additional isolates of six established *Ralstonia* species, including *R. insidiosa*, *R. mannitolilytica*, *R. pickettii*, *R. pseudosolanacearum*, *R. solanacearum,* and *R. syzygii*; (ii) 2 isolates of recently described species that were missing in the commercial Bruker database (IVD v12.0, MSP-11758), i.e., *R. chuxiongensis* and *R. wenshanensis*; and (iii) 11 isolates of the seven novel species detected in the present study (Table S3). After re-identification of the spectra of all 161 isolates examined using the commercial Bruker (IVD v12.0, MSP-11758) and the *in-house* database (RUO, MSP-5331), the identification scores allowed the identification of 124 isolates (77%) at the species level (Table S2). Twenty-six isolates that were identified at the genus level only were identified based on clustering of their mass spectra as *R. mannitolilytica* (23 isolates), *R. insidiosa* (2 isolates), or *R. wenshanensis* (1 isolate). Eleven isolates remained identified at the genus level only and represented novel species (Table S2). Of the latter, five *Ralstonia thomasii* isolates could not be identified at the species level because their mass spectra matched with the reference spectra of *R. thomasii*, *R. wenshanensis,* and *R. pickettii* with an identification score ≥2.3. With the commercial Bruker database (IVD v12.0, MSP-11758), four of these isolates were misidentified as *R. pickettii* (Table S2); one *Ralstonia holmesii* isolate was misidentified as *Ralstonia flatus* (Table S2).

### Genome analyses

The Illumina 150 bp paired-end reads of the 32 *Ralstonia* isolates yielded draft genomes with 13–145 contigs and estimated genome sizes between 4.48 and 5.78 Mbp (Table 1). The percentage G + C content ranged from 63.21% to 66.08% and the number of predicted CoDing Sequence (CDS) from 4,116 to 5,343. Pairwise dDDH and ANI values were calculated between the 32 new genomes and the genomes of the type strains of the 11 established *Ralstonia* species (Fig. S1). Species delineation based on the 70% dDDH (37, 51) and 95%–96% ANI thresholds (38, 52) allowed the identification of 12 isolates as *R. mannitolilytica*, 2 isolates as *R. wenshanensis*, and a single isolate each as *R. pickettii* and *R. chuxiongensis*. Comparison of genomic dDDH and ANI values among the remaining 16 genome sequences and toward the 11 established *Ralstonia* species revealed the presence of seven novel species (Fig. S1). A phylogenomic analysis based on 107 single-copy marker genes was well resolved, and the clusters delineated by dDDH and ANI formed monophyletic groups with high bootstrap support (Fig. 1). The proposed names for these seven novel species are shown in Table 1 and Table S1.

Classification of the 32 genomes generated in the present study in the Genome Taxonomy Database (39, 40) corroborated the species cluster delineation based on dDDH and ANI values. In addition, five novel species reported in the present study, i.e., *Ralstonia edaphis*, *Ralstonia flaminis*, *R. flatus*, *R. holmesii*, and *R. thomasii*, corresponded with an unnamed *Ralstonia* species in the GTDB database, thus matching hitherto unclassified public genomes (Table S4).

**TABLE 1** Genomes included in the present study

| Isolate | Assembly accession | Contigs | Size (Mbp) | G + C content (%) | CDS |
|---|---|---|---|---|---|
| R. pickettii K-288[T] | GCF_016466415 | 3[a] | 4.83 | 63.91 | 4,467 |
| R. pickettii R-38712 | GCA_958349765 | 71 | 5.62 | 63.57 | 5,294 |
| R. mannitolilytica LMG 6866[T] | GCF_905397375 | 36 | 4.82 | 65.81 | 4,454 |
| R. mannitolilytica LMG 8323 | GCA_958348635 | 91 | 4.59 | 65.85 | 4,242 |
| R. mannitolilytica LMG 18090 | GCA_958405575 | 48 | 5.08 | 65.77 | 4,694 |
| R. mannitolilytica LMG 18102 | GCA_958347605 | 28 | 4.83 | 65.86 | 4,457 |
| R. mannitolilytica R-1479 | GCA_958439005 | 59 | 4.92 | 65.99 | 4,574 |
| R. mannitolilytica R-76696 | GCA_958350155 | 43 | 4.95 | 65.99 | 4,642 |
| R. mannitolilytica R-76706 | GCA_958349945 | 29 | 4.52 | 65.93 | 4,156 |
| R. mannitolilytica R-76727 | GCA_958437825 | 145 | 4.87 | 66.08 | 4,553 |
| R. mannitolilytica R-77555 | GCA_958405885 | 28 | 4.71 | 65.83 | 4,323 |
| R. mannitolilytica R-77569 | GCA_958437905 | 58 | 5.24 | 65.73 | 4,931 |
| R. mannitolilytica R-77591 | GCA_958347435 | 63 | 5.30 | 65.40 | 4,934 |
| R. mannitolilytica R-77592 | GCA_958349955 | 104 | 4.80 | 65.62 | 4,461 |
| R. mannitolilytica R-82526 | GCA_958347445 | 31 | 4.76 | 66.04 | 4,435 |
| R. insidiosa CCUG 46789[T] | GCF_008801405 | 15 | 5.72 | 63.70 | 5,296 |
| R. solanacearum K60[T] | GCF_002251695 | 2 | 5.77 | 66.39 | 5,072 |
| R. pseudosolanacearum LMG 9673[T] | GCF_919586305 | 245 | 5.44 | 66.76 | 4,749 |
| R. syzygii subsp. syzygii LMG 10661[T] | GCA_919592095 | 348 | 3.99 | 66.50 | 3,840 |
| R. nicotianae RS[T] | GCF_018243235 | 2[a] | 5.61 | 67.10 | 4,772 |
| R. wenshanensis 56D2[T] | GCF_021173085 | 2[a] | 5.31 | 63.74 | 4,866 |
| R. wenshanensis LMG 18091 | GCA_958347615 | 17 | 5.48 | 63.65 | 5,079 |
| R. wenshanensis LMG 19087 | GCA_958415835 | 27 | 5.41 | 63.71 | 4,978 |
| R. chuxiongensis 21YRMH01-3[T] | GCF_024158925 | 20 | 5.61 | 63.47 | 5,228 |
| R. chuxiongensis LMG 32966 | GCA_958404865 | 88 | 5.78 | 63.21 | 5,342 |
| R. mojiangensis 21MJYT02-10[T] | GCF_023955645 | 14 | 5.60 | 63.56 | 5,183 |
| R. soli 21MJYT02-11[T] | GCF_023955655 | 31 | 5.73 | 64.12 | 5,241 |
| R. condita sp. nov. LMG 7141[T] | GCA_958405655 | 17 | 4.48 | 64.41 | 4,117 |
| R. edaphis sp. nov. LMG 6871[T] | GCA_958349005 | 19 | 5.29 | 64.46 | 4,843 |
| R. edaphis sp. nov. LMG 19089 | GCA_958347695 | 13 | 5.32 | 64.49 | 4,942 |
| R. edaphis sp. nov. R-16034 | GCA_958350185 | 18 | 5.33 | 64.49 | 4,897 |
| R. flaminis sp. nov. LMG 18101[T] | GCA_958415495 | 47 | 5.68 | 63.33 | 5,215 |
| R. flatus sp. nov. LMG 32965[T] | GCA_958439105 | 25 | 5.61 | 64.19 | 5,163 |
| R. flatus sp. nov. R-20233 | GCA_958438995 | 19 | 5.34 | 64.40 | 4,924 |
| R. flatus sp. nov. R-77567 | GCA_958437745 | 27 | 5.19 | 64.39 | 4750 |
| R. holmesii sp. nov. LMG 32967[T] | GCA_958347595 | 22 | 5.43 | 63.69 | 5,030 |
| R. holmesii sp. nov. LMG 18093 | GCA_958415225 | 27 | 5.37 | 63.62 | 4,898 |
| R. holmesii sp. nov. LMG 18096 | GCA_958405615 | 34 | 5.52 | 63.65 | 5,067 |
| R. psammae sp. nov. LMG 19083[T] | GCA_958405935 | 58 | 5.51 | 63.86 | 4,999 |
| R. thomasii sp. nov. LMG 18095[T] | GCA_958405895 | 39 | 5.04 | 63.80 | 4,673 |
| R. thomasii sp. nov. LMG 7143 | GCA_958348425 | 26 | 4.91 | 63.90 | 4,595 |
| R. thomasii sp. nov. R-6138 | GCA_958439175 | 39 | 4.91 | 63.90 | 4,569 |
| R. thomasii sp. nov. R-77560 | GCA_958405925 | 32 | 5.08 | 63.85 | 4,756 |

[a]Complete genome.

## Antimicrobial resistance and virulence genes

The 32 genome sequences generated in the present study and the genomes of the type strains of 11 established *Ralstonia* species were searched for antimicrobial resistance (CARD) and virulence (VFDB) genes using ABRicate (Fig. 2; Table S5). Antimicrobial resistance genes were detected in all genomes except *R. syzygii* LMG 10661[T]. Most resistance genes encoded aminoglycoside or beta-lactam resistance (Table S5), and beta-lactam resistance genes were detected in all genomes except those of

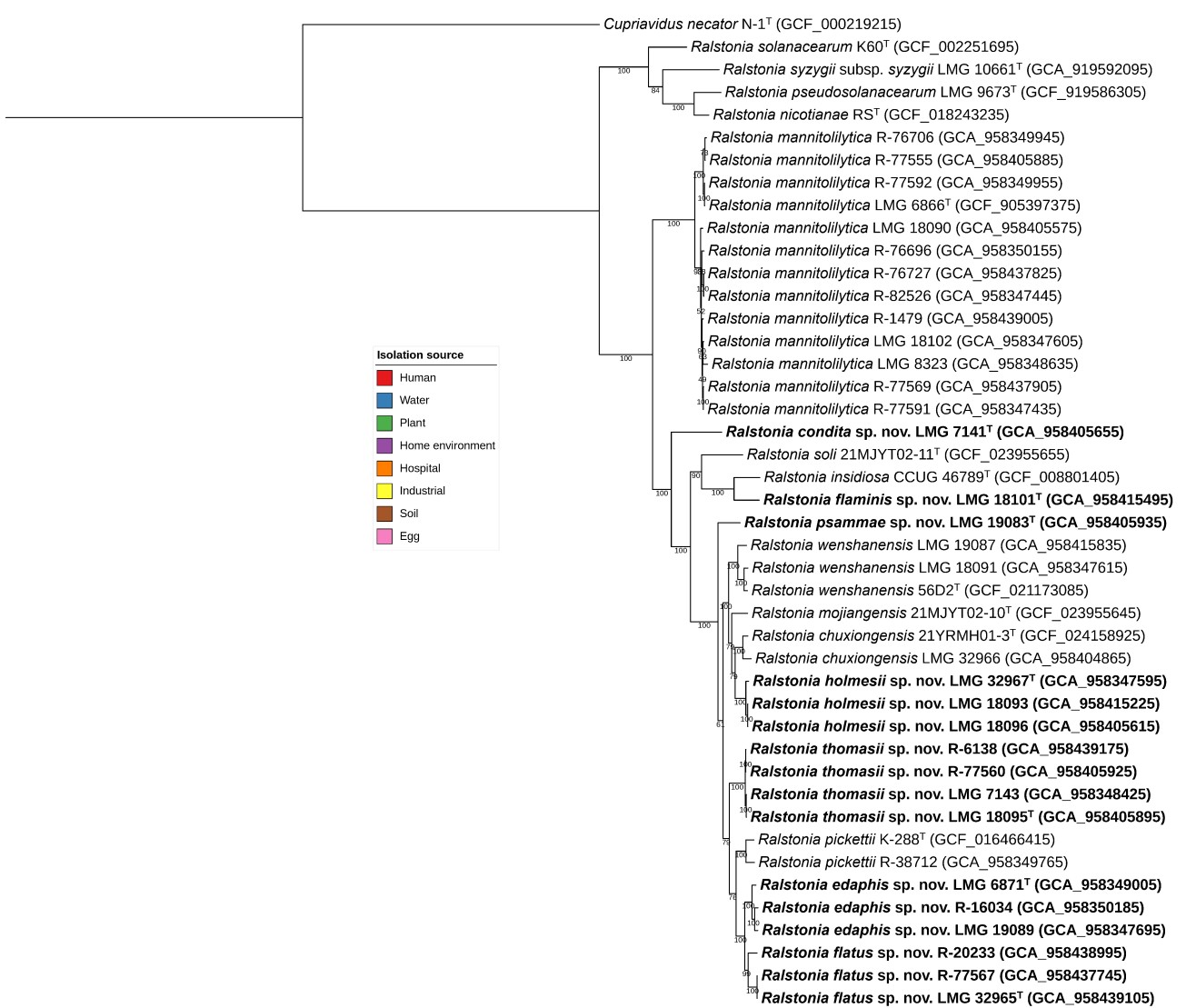

**FIG 1** Phylogenomic tree showing the phylogenetic relationship of genomes in the present study toward type strains of established *Ralstonia* species. BcgTree (42) was used to extract the amino acid sequence of 107 single-copy core genes and perform a partitioned maximum-likelihood analysis on the concatenated sequences (36,742 positions). *Cupriavidus necator* N-1^T was used as an outgroup. The percentage of replicate trees in which the associated taxa clustered together in the bootstrap test (1,000 replicates) are shown next to the branches. Visualization and annotation of the tree were performed using iTOL (43). Assembly accession numbers are given between parentheses. Bar, 0.1 changes per nucleotide position.

plant-pathogenic species (Fig. 2). Only a few virulence factors were detected, i.e., *flgG* and *bopC* (Fig. 2; Table S5).

## MIC analyses

A total of 30 isolates were analyzed for antimicrobial susceptibility, including a single reference isolate each of *R. pickettii* and *R. insidiosa*, five clinical isolates of *R. mannitolilytica*, one and two isolates of the recently described *R. chuxiongensis* and *R. wenshanensis*, respectively, and isolates of the novel clinical and environmental *Ralstonia* species reported in the present study (Table S1). Generally, the resistance patterns to ciprofloxacin, tigecycline, and trimethoprim/sulfamethoxazole differed from those toward other antibiotics, as most isolates were found susceptible (Fig. 3). High resistance patterns were observed against most beta-lactam antibiotics examined, except for piperacillin,

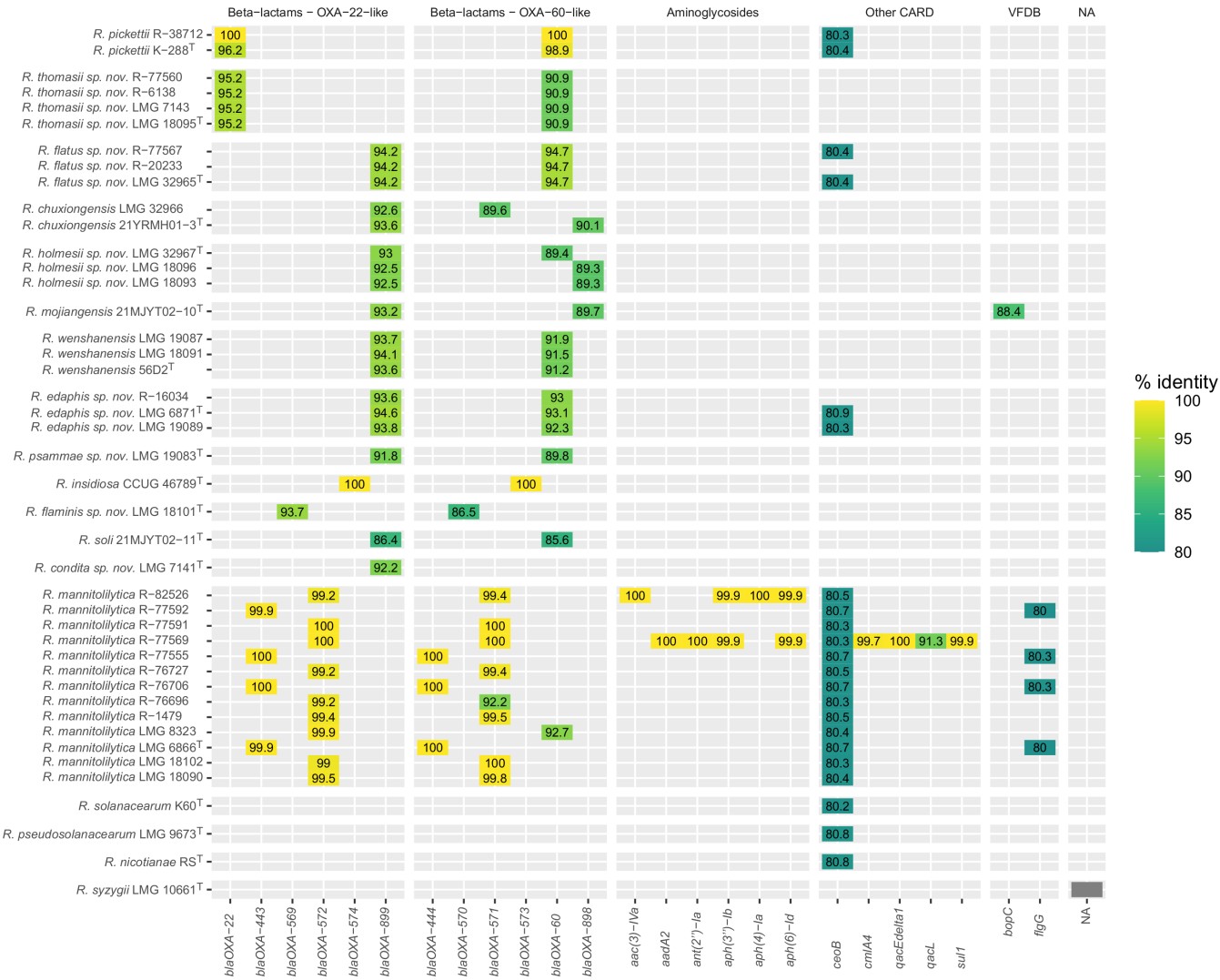

**FIG 2** Antimicrobial resistance and virulence genes found using ABRicate with CARD and VFDB databases. NA, no hits. Heatmap color scale represents percentage nucleotide identity with the reference genes from CARD and VFDB. Gene product and resistance mechanism are given in Table S5.

piperacillin-tazobactam, and cefepime, for which mixed resistance patterns were found. Most isolates exhibited resistance to ceftazidime, ceftazidime/avibactam, and ceftolozane/tazobactam, except for *Ralstonia condita* LMG 7141[T] and *R. mannitolilytica* R-16028, which were susceptible to all three cephalosporins (Fig. 3). The former isolate was susceptible to most beta-lactam antibiotics examined. Finally, most isolates exhibited resistance to colistin, amikacin, and tobramycin (Fig. 3).

## Phenotypic and biochemical characterization

A total of 19 isolates were selected for phenotypical and biochemical characterization, including a single reference isolate each of *R. pickettii*, *R. insidiosa*, *R. chuxiongensis*, and *R. wenshanensis*, 7 isolates of *R. mannitolilytica*, and type strains of the novel *Ralstonia* species reported in the present study (Table S1). Cells of each of the 19 isolates examined were rod shaped and occurred alone, in pairs, or in short chains. The cells were approximately 0.5 µm in diameter and between 1.5 and 2 µm long. After 48 h of incubation on TSA, colonies appeared opaque, circular, convex with smooth surface, and with wavy or smooth margins with no distinct protrusions.

All grew on MacConkey, Drigalski, and R2A agar (24 h, 28°C). There was no growth on TSA in anaerobic conditions; anaerobic growth on TSA supplemented with KNO₃ was a

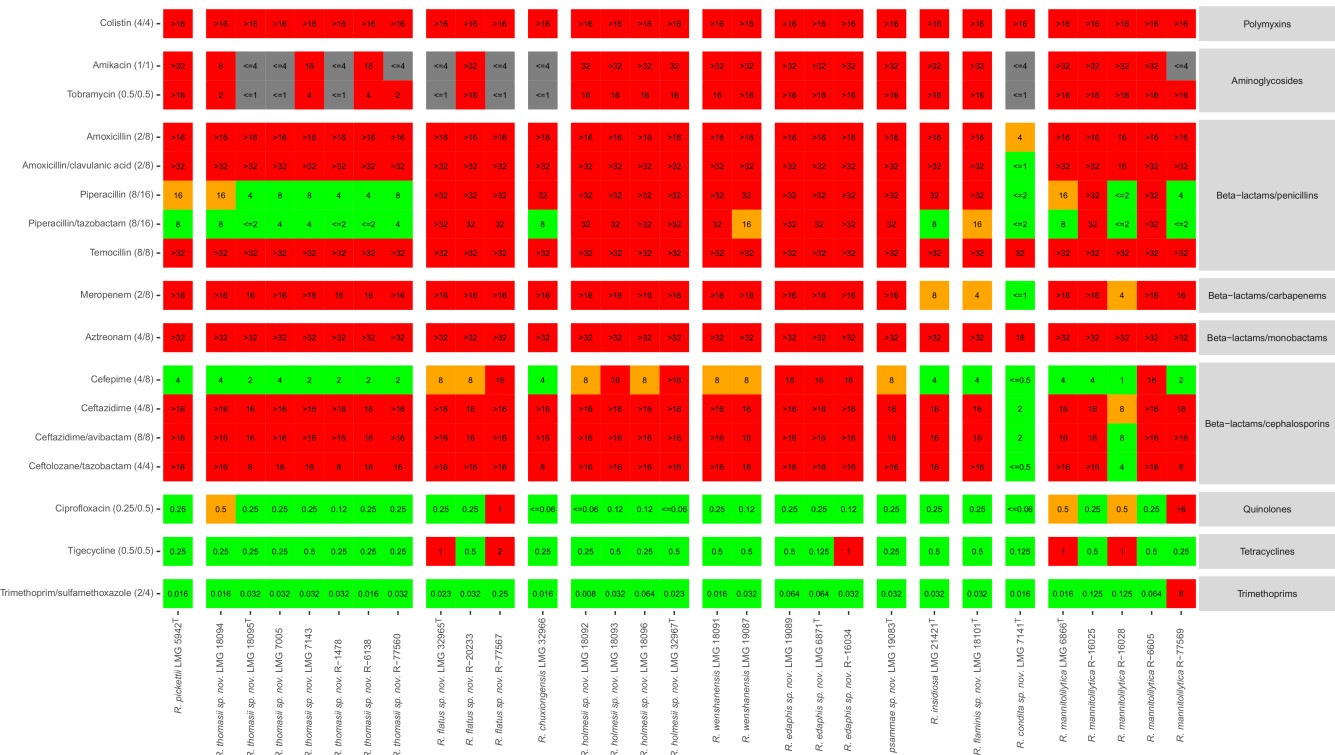

**FIG 3** Antimicrobial resistance patterns of 30 *Ralstonia* isolates. MIC values for aminoglycoside, beta-lactam, fluoroquinolone, and tetracycline antibiotics were interpreted using EUCAST PK/PD breakpoints. Breakpoints for colistin were based on those from EUCAST *Pseudomonas* spp., and breakpoints for trimethoprim/sulfamethoxazole were based on those from EUCAST *Acinetobacter* spp.. Interpretation is summarized visually: red, "resistant"; orange, "susceptible, increased exposure"; green, "susceptible, normal exposure." In case of aminoglycoside antibiotics, gray indicates that the tested concentration range was inappropriate to conclude susceptible or resistant. Tested concentrations were 1–16 µg/mL for colistin, tobramycin, amoxicillin, meropenem; 4–32 µg/mL for amikacin; 2–32 µg/mL for piperacillin and temocillin; 0.5–32 µg/mL for aztreonam; 0.5–16 µg/mL for cefepime and ceftazidime; 1/2–32/2 µg/mL for amoxicillin/clavulanic acid; 2/4–32/4 µg/mL for piperacillin/tazobactam; 0.5/4–16/4 for ceftazidime/avibactam and ceftolozane/tazobactam; 0.06–8 µg/mL for ciprofloxacin; 0.016–256 (E-test) for tigecycline; and 0.002–32 (E-test) for trimethoprim/sulfamethoxazole.

species-specific characteristic (Fig. S2). All isolates hydrolyzed gelatin, and none hydrolyzed starch. None of the isolates, except *R. chuxiongensis* LMG 32966, showed DNase activity. All isolates were oxidase and catalase positive except for *R. mannitolilytica* LMG 8323 (no catalase activity) and *R. mannitolilytica* R-77569 (no oxidase activity). Nitrate reduction, casein hydrolysis, and hemolytic activity were strain dependent (Fig. S2). Few isolates grew on cetrimide agar. Optimal growth was observed between 15°C and 40°C, except for *R. mannitolilytica* isolates, which grew optimally between 20°C and 45°C. Growth at different temperatures did not differ when tested on TSA or NA. All isolates grew in the presence of up to 2% NaCl and at pH 6–8. No growth was observed at 6%–10% NaCl or at pH 4–5. Acid was oxidatively produced from D-glucose by all isolates except *R. condita* LMG 7141[T]. None of the isolates fermented D-glucose. Acid production from D-mannitol and D-arabitol was observed for a few isolates only (Fig. S2).

## DISCUSSION

*Ralstonia* bacteria have been reported as opportunistic pathogens in a variety of human infections (19). An increasing prevalence of *Ralstonia* infections in persons with cystic fibrosis has been attributed to improved diagnostics, improved awareness, and a genuine increase in case numbers (4, 18, 23, 53). MALDI-TOF mass spectrometry is now commonly used for the diagnosis of bacterial infections, yet the commercial Bruker database (IVD v12.0, MSP-11758) holds reference spectra of only nine *R. insidiosa* isolates, nine *R. pickettii* isolates, five *R. mannitolilytica* isolates, one *R. syzygii* isolate, and two

*Ralstonia* sp. isolates (Table S3). In the present study, an initial analysis of 161 human clinical and environmental isolates using the commercial Bruker database (IVD v12.0, MSP-11758) yielded confident species-level identification (Bruker scores ≥ 2.3) for only 62 isolates (39%), including 4 isolates that proved misidentified subsequently (Table S2). Preliminary analyses of public and own whole-genome sequences (data not shown) suggested the existence of several novel human clinical and environmental *Ralstonia* species (23). We therefore analyzed the diversity of mass spectra generated in the present study using global pattern similarities and unique spectral features (Table S2) and selected 32 isolates that were not reliably identified at the species level for whole-genome sequence analyses. Overall genomic relatedness indices identified 12 isolates as *R. mannitolilytica*, 2 isolates (54) as *R. wenshanensis*, and a single isolate each as *R. pickettii* and *R. chuxiongensis*, and revealed the presence of 7 novel *Ralstonia* species, among which 3 and 4 that were isolated from CF and other human clinical samples, respectively (Table S1). These identification results were subsequently used to select 32 isolates that represented poorly identified or new *Ralstonia* species for creating additional MSPs (Table S3).

Reidentification of all MALDI-TOF mass spectra with the updated *in-house* database yielded a confident species level identification for 124 isolates (77%) (Table S2) demonstrating that the identification of *Ralstonia* isolates can be improved drastically through the construction of a more complete reference database (55, 56). Five *R. thomasii* isolates could not be identified at the species level because their mass spectra matched with reference spectra of different *Ralstonia* species; four of these isolates were misidentified as *R. pickettii* with the commercial Bruker database (IVD v12.0, MSP-11758) (Table S2). The mass spectra of the latter four isolates matched the mass spectrum of *R. pickettii* CCUG 30895 with an identification score ≥2.3, and none of the mass spectra of isolates identified as *R. pickettii* in the present study matched the mass spectrum of *R. pickettii* CCUG 30895, suggesting that this MSP may have been misidentified in the commercial Bruker database (IVD v12.0, MSP-11758). Finally, *R. holmesii* LMG 18093 was misidentified as *R. flatus* with an identification score ≥2.3. The addition of more reference spectra of poorly represented novel *Ralstonia* species to the identification database may further improve MALDI-TOF MS-based identification results.

Human clinical bacteria now known as *Ralstonia* species were first classified as *Pseudomonas pickettii* (8). Subsequently, Riley and Weaver identified the unnamed taxon CDC group Va-2 as *P. pickettii* (57) through detailed biochemical analyses. King et al. (58) and Pickett and Greenwood (59) proposed classifying CDC group Va-1 as *P. pickettii* as well, again primarily on the basis of extensive biochemical testing. Another pseudomonad first isolated from a nosocomial outbreak caused by contaminated autoclaved fluids at St Thomas' Hospital, London, UK (60) was referred to as "*Pseudomonas thomasii*" (this name was not validly published). Additional nosocomial outbreaks caused by "*P. thomasii*"-contaminated distilled water, chlorhexidine solutions, or other hospital supplies, and sporadic cases were reported (61–64). While early taxonomic work subsequently also considered "*P. thomasii*" as yet another biovar of *P. pickettii* (58, 65), King et al. (58) stressed the taxonomic heterogeneity of the *P. pickettii*–"*P. thomasii*" cluster, and Costas et al. (62) emphasized the taxonomic distinctiveness of *P. pickettii* and "*P. thomasii*" on the basis of whole-cell protein electrophoretic analyses and preliminary DNA–DNA hybridization analyses. Finally, De Baere et al. (7) classified one of the isolates of the St Thomas' Hospital outbreak (60) and several more recent human clinical isolates in the species *R. mannitolilytica*. The large majority of human clinical isolates of the present study were identified as *R. mannitolilytica* (Table S1), confirming its importance as the most prevalent human clinical *Ralstonia* species (5, 7, 17–19, 24). Twelve isolates were confirmed as *R. pickettii,* and the remaining human clinical isolates represented five novel *Ralstonia* species (Table S1).

The present study included several additional "*P. thomasii*" isolates from historical strain collections. Strain R-288 (= NCTC 10893) (58, 62) is another isolate of the St Thomas' Hospital outbreak (60), and was confirmed as *R. mannitolilytica* (Table S1), as

was R-24685 (= CL43/89) (62), an isolate of an outbreak in a special care baby unit in St. Bartholomew's Hospital, London, UK (62, 63). Strains LMG 18094 (= CL24/74) (58) and LMG 18095$^T$ (= CL78/74) (58) represent sporadic cases of "*P. thomasii*" infections in St Thomas' Hospital (58). Results of the present study demonstrated that these belong to a novel clinical *Ralstonia* species, which we propose to name *R. thomasii*, along with several other historical CDC group Va-1 isolates (58, 59, 66) and some recent clinical isolates (Table S1). Similarly, strains LMG 18093 (= 118700) (58) and LMG 18096 (= CL605/72) (58) also represent sporadic cases of "*P. thomasii*" infections in St Thomas' Hospital (58) and were identified in the present study as yet another novel clinical *Ralstonia* species, *R. holmesii,* along with LMG 18092 (= Pickett K-615) (58) and a recent river water isolate (Table S1). Strain LMG 7141$^T$ (=Pickett K-1303), a CDC group Va-1 strain (59) represents the novel clinical *Ralstonia* species *R. condita*. Finally, the novel species *R. flaminis* and *R. flatus* include recent isolates from sputum of persons with CF and blood samples (Table S1).

The phenotypic and biochemical characteristics recorded in the present study (Fig. S2) were generally consistent with earlier data with one notable exception (6–9, 14, 57). Although we tested several experimental procedures (data not shown), we failed to detect acidification of D-arabitol or D-mannitol by *R. mannitolilytica* (seven isolates were tested), a key diagnostic feature reported earlier (7), while other isolates acidified both carbon sources (Fig. S2). This discrepancy may be explained by different test procedures (16, 65).

*Ralstonia* species are generally multi-resistant to antibiotics (17). This was largely confirmed in the present study upon MIC analysis of five recent *R. mannitolilytica* isolates and of isolates representing novel *Ralstonia* species (Fig. 3) and may be supported by the detection of three genes encoding subunits of multi-drug efflux pumps (Fig. 2; Table S5). A first, *ceoB*, encodes the cytoplasmic membrane component of the CeoAB-OpcM efflux pump in *Burkholderia cenocepacia*, which provides resistance against fluoroquinolone and aminoglycoside antibiotics (67). The remaining two, i.e., *qacEdelta1* and *qacL*, encode subunits of the qac efflux pump, which provides resistance against quaternary ammonium component antiseptics and intercalating dyes (68–70). One *R. mannitolilytica* isolate (R-16028), however, showed lower *in vitro* resistance patterns to beta-lactam antibiotics indicating that not all clinical *R. mannitolilytica* isolates are multi-resistant. The latter isolate was identified as *R. mannitolilytica* based on MALDI-TOF MS analysis only (Table S2).

Trimethoprim/sulfamethoxazole and fluoroquinolone antibiotics are considered the best treatment options for *Ralstonia* infections (17, 23). In line with this, the highest *in vitro* susceptibility was found for trimethoprim/sulfamethoxazole, ciprofloxacin, and tigecycline (17, 23, 71–73). We detected little or no *in vitro* activity for amikacin, tobramycin, colistin, or some of the beta-lactam antibiotics (amoxicillin, temocillin, aztreonam) studied. A single isolate, i.e., *R. mannitolilytica* R-77569, was categorized as resistant to trimethoprim/sulfamethoxazole (Fig. 3). Its genome comprised the *sul1* gene (Fig. 2), which encodes a dihydropteroate synthase that restores folic acid metabolism and thus counteracts inhibition by the sulfonamide antibiotic (74). Aminoglycoside resistance commonly occurs through enzymatic inactivation (75), although lipopolysaccharide modifications may also contribute to resistance (76, 77). Only two isolates, i.e., *R. mannitolilytica* R-82526 and R-77569, encoded aminoglycoside-modifying enzymes (Fig. 2). Variable *in vitro* susceptibility patterns were observed for beta-lactam antibiotics. All isolates, except *R. condita* LMG 7141$^T$, were resistant to most beta-lactam antibiotics analyzed (Fig. 3) and encoded a gene for a class D beta-lactamase of both the OXA-22-like and OXA-60-like subfamily (which was absent in LMG 7141$^T$) (Fig. 2). The hydrolytic spectrum of the OXA-22-like and OXA-60-like class D beta-lactamases does, however, not fully explain the observed resistance to the beta-lactam antibiotics (78, 79). The genome of *R. condita* LMG 7141$^T$ was atypical as only bla$_{OXA-899}$, a class D beta-lactamase of the OXA-22-like subfamily, was detected. OXA-gene sequences of established *Ralstonia* species had much higher percentage identity values with OXA-gene sequences in the

CARD database compared to those of the novel *Ralstonia* species reported in the present study (Fig. 2). Finally, some *Ralstonia* isolates showed less resistance to piperacillin, piperacillin–tazobactam, and cefepime (Fig. 3).

In conclusion, the present study demonstrated that a commercial MALDI-TOF MS identification database can be tailored to improve the identification of *Ralstonia* species. It also confirms the existence of five novel *Ralstonia* species from human clinical sources and the hospital environment. An analysis of MIC values demonstrated that the novel *Ralstonia* species were generally multi-resistant but susceptible to trimethoprim/sulfa-methoxazole, ciprofloxacin, and tigecycline.

## Novel species descriptions

### Description of Ralstonia condita sp. nov.

*Ralstonia condita* (con.di'ta. L. adj. *condita* hidden, because the distinct taxonomic status of this CDC group Va-1 strain remained hidden for more than half a century).

The phenotypic characteristics are as described above and presented in Fig. S2. The type strain is LMG 7141$^T$ (= CCUG 77163$^T$) and was isolated from the hospital environment in the USA. Its G + C content is 64.41%. The 16S rRNA and whole-genome sequence of LMG 7141$^T$ are publicly available through the accession numbers OY696110 and GCA_958405655, respectively.

### Description of Ralstonia thomasii sp. nov.

*Ralstonia thomasii* [tho.ma'si.i. N.L. gen. n. *thomasii* from Thomas, referring to the Saint Thomas Hospital (London, UK) where an outbreak caused by these bacteria was reported].

The phenotypic characteristics are as described above and presented in Fig. S2. Isolated from human clinical samples and the hospital environment. The type strain is LMG 18095$^T$ (= CCUG 38763$^T$). Its G + C content is 63.80%. The 16S rRNA and whole-genome sequence of LMG 18095$^T$ are publicly available through the accession numbers OY696106 and GCA_958405895, respectively.

### Description of Ralstonia holmesii sp. nov.

*Ralstonia holmesii* (hol.me'si.i. N.L. gen. n. *holmesii* of Holmes, named after Barry Holmes, an English microbiologist, for his many contributions to the taxonomy and diagnosis of human clinical bacteria).

The phenotypic characteristics are as described above and presented in Fig. S2. Isolated from human clinical and environmental samples. The type strain is LMG 32967$^T$ (= CCUG 77162$^T$). Its G + C content is 63.69%. The 16S rRNA and whole-genome sequence of LMG 32967$^T$ are publicly available through the accession numbers OY696112 and GCA_958347595, respectively.

### Description of Ralstonia flatus sp. nov.

*Ralstonia flatus* (fla'tus. L. gen. n. *flatus* from a breath).

The phenotypic characteristics are as described above and presented in Fig. S2. Isolated from human clinical and food samples. The type strain is LMG 32965$^T$ (= CCUG 77161$^T$). Its G + C content is 64.19%. The 16S rRNA and whole-genome sequence of LMG 32965$^T$ are publicly available through the accession numbers OY696111 and GCA_958439105, respectively.

### Description of Ralstonia flaminis sp. nov.

*Ralstonia flaminis* (fla'mi.nis. L. gen. n. *flaminis* from an exhalation).

The phenotypic characteristics are as described above and presented in Fig. S2. The type strain is LMG 18101$^T$ (= CCUG 38754$^T$) and was isolated from the sputum of a CF patient (Canada). Its G + C content is 63.33%. The 16S rRNA and whole-genome sequence of LMG 18101$^T$ are publicly available through the accession numbers OY696107 and GCA_958415495, respectively.

### Description of Ralstonia edaphis sp. nov.

*Ralstonia edaphis* (e.da'phis. Gr. neut. n. *edaphos* soil; N.L. gen. n. *edaphis*, from soil).

The phenotypic characteristics are as described above and presented in Fig. S2. Isolated from soil samples. The type strain is LMG 6871$^T$ (= CCUG 18841$^T$). Its G + C content is 64.46%. The 16S rRNA and whole-genome sequence of LMG 6871$^T$ are publicly available through the accession numbers OY696109 and GCA_958349005, respectively.

### Description of Ralstonia psammae sp. nov.

*Ralstonia psammae* (psam'mae. Gr. fem. n. *psamme* sand; N.L. gen. n. *psammae*, from sand, referring to the sandy soil isolation source).

The phenotypic characteristics are as described above and presented in Fig. S2. Isolated from soil samples. The type strain is LMG 19083$^T$ (= CCUG 77164$^T$). Its G + C content is 63.86%. The 16S rRNA and whole-genome sequence of LMG 19083$^T$ are publicly available through the accession numbers OY696108 and GCA_958405935, respectively.

## ACKNOWLEDGMENTS

We thank the Oxford Genomics Centre at the Welcome Centre for Human Genetics (funded by Wellcome Trust grant reference 203141/Z/16/Z) for the generation and initial processing of the sequencing data. We thank Microbial Genome Sequencing Center (MiGS) for the high-quality genome sequencing data. We thank Evelien De Canck, Margo Cnockaert, and all lab technicians from BCCM/LMG for their technical assistance. The computational resources (Stevin Supercomputer Infrastructure) and services used in this work were provided by the VSC (Flemish Supercomputer Center), funded by Ghent University, FWO, and the Flemish Government—Department EWI. This research was carried out in part using infrastructure funded by EMBRC Belgium—FWO international research infrastructure I001621N.

Conceptualization: S.S., C.P., A.D.W., and P.V.; data curation: S.S., C.P., A.D.W., T.S., J.J.L., and P.V.; formal analysis: S.S., C.P., A.D.W., and P.V.; funding acquisition: I.W., D.P, J.J.L., and P.V.; methodology: S.S., C.P., A.D.W., A.M., K.V., D.P., and P.V.; project administration: S.S., C.P., A.D.W., and P.V.; software: A.D.W. and C.P.; supervision: P.V.; visualization: C.P.; writing—original draft preparation: S.S., C.P., and P.V.; writing—review and editing: S.S., C.P., J.J.L., and P.V. All authors read and approved the final manuscript.

The authors declare that they have no known competing financial interests or personal relationships that could have appeared to influence the work reported in this paper.

## AUTHOR AFFILIATIONS

¹Laboratory of Microbiology, Department of Biochemistry and Microbiology, Ghent University, Gent, Belgium
²National Reference Center for *Burkholderia cepacia* complex, La Plata, Belgium
³BCCM/LMG Bacteria Collection, Laboratory of Microbiology, Department of Biochemistry and Microbiology, Ghent University, Ghent, Belgium
⁴Department of Microbiology and Infection Control, Vrije Universiteit Brussel (VUB), Universitair Ziekenhuis Brussel (UZ Brussel), Brussels, Belgium
⁵Department of Pediatrics, University of Michigan Medical School, Ann Arbor, Michigan, USA

## PRESENT ADDRESS

Astrid Muyldermans, Department of Laboratory Medicine, Medical Microbiology, AZ Sint-Jan, Brugge, Belgium

## AUTHOR ORCIDs

Stephanie Steyaert  http://orcid.org/0000-0002-6953-7047
Charlotte Peeters  http://orcid.org/0000-0002-1891-4869
Anneleen D. Wieme  http://orcid.org/0000-0003-4503-4063
Ingrid Wybo  https://orcid.org/0000-0002-6217-1695
John J. LiPuma  http://orcid.org/0000-0003-4033-7794
Peter Vandamme  http://orcid.org/0000-0002-5581-7937

## DATA AVAILABILITY

All genome sequences determined in the present study are available at the European Nucleotide Archive (ENA) under the study accession numbers PRJEB43925 and PRJEB63170.

## ADDITIONAL FILES

The following material is available online.

### Supplemental Material

**Supplemental figures (Spectrum04021-23-S0001.pdf).** Fig. S1 and S2.
**Supplemental tables (Spectrum04021-23-S0002.xlsx).** Tables S1-S5.

### Open Peer Review

**PEER REVIEW HISTORY (review-history.pdf).** An accounting of the reviewer comments and feedback.

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
