## [Reviewer comments · Microbiology Spectrum]

Microbiology Spectrum

Novel *Ralstonia* species from human infections: improved matrix-assisted laser desorption/ionization time-of-flight mass spectrometry-based identification and analysis of antimicrobial resistance patterns

Stephanie Steyaert, Charlotte Peeters, Anneleen Wieme, Astrid Muyldermans, Kristof Vandoorslaer, Theodore Spilker, Ingrid Wybo, Denis Piérard, John LiPuma, and Peter VANDAMME

Corresponding Author(s): Charlotte Peeters, Universiteit Gent

Review Timeline:

Submission Date:	November 28, 2023
Editorial Decision:	January 8, 2024
Revision Received:	February 15, 2024
Editorial Decision:	March 2, 2024
Revision Received:	March 6, 2024
Accepted:	March 13, 2024

Editor: Silvia Cardona

Reviewer(s): Disclosure of reviewer identity is with reference to reviewer comments included in decision letter(s). The following individuals involved in review of your submission have agreed to reveal their identity: Mikel Joseba Urrutikoetxea (Reviewer #1); Alejandra Bosch (Reviewer #2)

Transaction Report:

DOI: <https://doi.org/10.1128/spectrum.04021-23>

Re: Spectrum04021-23 (Improved matrix-assisted laser desorption/ionization time-of-flight mass spectrometry-based identification and antimicrobial resistance patterns of *Ralstonia* recovered from human infections)

Dear Dr. Charlotte Peeters:

Thank you for submitting your manuscript to Microbiology Spectrum. Your article has been reviewed by three experts in the field. While reviewers found your work merits publication, they have provided comments that, if addressed, will improve the quality of the work substantially.

Their recommendations are provided below.

Revision Guidelines

Sincerely,
Silvia Cardona
Editor
Microbiology Spectrum

Reviewer #1 (Comments for the Author):

Several years ago Bruker changed the score classification and this species and genus level identification classification is no longer in place (lines 91-94) and the manufacturer classifies them as: (2.00 - 3.00) High Confidence Identification, (1.70 - 1.99) Low Confidence Identification and (0.00 - 1.69) No Organism Identification Possible. How would the performance vary using the

current definitions for both the commercial and the in-house library.

Why were the *Pseudomonas* spp. breakpoints for Aminoglycosides used instead of the PK/PD ones?

The use of species (as in *Ralstonia* species) and *Ralstonia* sp. instead of *Ralstonia* spp. should be reviewed through the whole manuscript (as in line 285 for example).

The same situation occurs with *Pseudomonas* spp. and *Acinetobacter* spp. (lines 175-176 and 499-500).

Reviewer #2 (Comments for the Author):

In the work titled "Improved matrix-assisted laser desorption/ionization time-of-flight mass spectrometry-based identification and antimicrobial resistance patterns of *Ralstonia* recovered from human infections" by Stephanie Steyaerta et al., presents an interesting analysis of a collection of 161 clinical and environmental *Ralstonia* isolates. The authors subjected this population to a MALDI-TOF dereplication and identification using the commercial Bruker database (IVD V12.0, MSP-11758). The identification scores allowed the confident identification of only 62 isolates (39%) down to the species level.

Based on this result they selected 32 representative isolates for which no confident MALDI-TOF MS species level identification was achieved, to accomplish a Whole-genome sequence analysis. The study revealed the presence of seven novel *Ralstonia* species from clinical and environmental sources. Using these novel sequences and the sequences of other 6 known species they developed an improved in-house MALDI TOF database (RUO, MSP-5331). A reanalysis of the original *Ralstonia* population, using this updated MALDI-TOF database increased the percentage of isolates with confident species level identification up to 77%.

The authors analyzed the antimicrobial resistance genes in 43 genomes comprising the novel genomes reported in this investigation and the genomes of 11 established *Ralstonia* species demonstrating a broadly distributed beta-lactam resistance determinants within the genus *Ralstonia*. In addition, they determined the antimicrobial susceptibility of selected human clinical and environmental *Ralstonia* isolates against 17 antimicrobial agents. Most of the novel *Ralstonia* species were multi-resistant, but most of the population is still susceptible to trimethoprim /sulfamethoxazole, ciprofloxacin and tigecycline.

Ralstonia species, can cause significant disease in certain at-risk patient groups, including those with cystic fibrosis. Challenges in accurate clinical laboratory identification arise due to frequent misidentification with closely related bacteria, the dynamic taxonomy within this genus, their notable multidrug-resistant profiles, and species-specific variations in antimicrobial resistance. Consequently, conducting systematic research aimed at enhancing identification methods and comprehensively analyzing antimicrobial resistance becomes imperative. Hence, the findings presented here represent a significant leap forward in our understanding of this genus."

Ralstonia species, can cause significant disease in certain at-risk patient groups, including those with cystic fibrosis. Challenges in accurate clinical laboratory identification arise due to frequent misidentification with closely related bacteria, the dynamic taxonomy within this genus, their notable multidrug-resistant profiles, and species-specific variations in antimicrobial resistance. Consequently, conducting systematic research aimed at enhancing identification methods and comprehensively analyzing antimicrobial resistance becomes imperative. Hence, the findings presented here substantially advance our understanding of the *Ralstonia* genus.

Minor concerns:

Title: as MALDI-TOF-MS has also been applied in the detection of drug-resistance mechanisms and particularly the detection of resistance against β -Lactam antibiotics, I would propose:

Improved matrix-assisted laser desorption/ionization time-of-flight mass spectrometry-based identification and analysis of antimicrobial resistance patterns of *Ralstonia* recovered from human infections

However, from my perspective I believe this title does not fully capture the main contributions of this manuscript, which I believe involve, the improvement of the *Ralstonia* MALDI-TOF MS-based identification primarily through the incorporation of seven novel *Ralstonia* species into the commercial database, and the analysis of antimicrobial resistance patterns of *Ralstonia* recovered from human infections. Wouldn't it be important to mention in the title that this investigation describes seven novel *Ralstonia* species? The improvement in MALDI-TOF-MS identification in this work does not lie on the MALDI-TOF MS methodology itself but rather on the development of an updated database.

Results section:

Lines 199 to 222: Given that the number of isolates selected for WGS identification was 32 (indicated in Supplementary table S2) it coincides with the number of isolates used to update the in-house MALDI-TOF database (presented in Supplementary

table S3), it might be advisable to alert the reader in line 199: "The in-house MALDI-TOF MS database was subsequently updated to include MSPs of (i) 19 isolates of 6 established *Ralstonia* species, including *R. insidiosa*, *R. mannitolilytica*, *R. pickettii*, *R. pseudosolanacearum*, *R. solanacearum*, and 3 subspecies of *R. syzygii*, (ii) 2 isolates of recently described species that were missing in the commercial Bruker database (IVD V12.0, MSP-11758), and (iii) 11 isolates of the 7 novel species detected in the present study (Supplementary table S3).

Discussion section

Line 314. The authors could include in the discussion whether the identification at the genus level or the misidentification of certain isolates could be attributed to the low availability of certain isolates mainly among the newly incorporated species (e.g. for *R. flaminis* and *R. condita* only one representative isolate of each species was available to be incorporated into the database).

Reviewer #3 (Comments for the Author):

This is a very interesting article on the identification of *Ralstonia* species. *Ralstonia* spp are pathogens for humans or plants, depending on the species concerned, and are considered to be emerging in cystic fibrosis patients. Epidemiological data is lacking because these pathogens are rarely isolated from human samples and identification techniques are not reliable for the species. Using a collection of 161 isolates, the authors highlight the limitations of the commercially available MALDI-TOF MS Bruker database for accurate species identification and suggest that implementation of the existing database would improve results. In addition, the authors describe 7 novel species within the genus, 4 of which were detected from human samples. Finally, the authors report antimicrobial resistance profiles of 30 isolates belonging to 12 species, providing new data.

Major comments :

- The reference method used to assess the performance of MALDI-TOF MS to identify isolates is not provided (apart from reference strains of established species, or WGS for 32 isolates). For the other isolates, which identification method was used?
- The authors suggest that the isolate identifications used in the Bruker IVD database may be erroneous (this has already been described for other bacterial genera). However, they still used a combination of the Bruker IVD database and their own database to identify isolates. This could therefore lead to erroneous identifications. To build an accurate identification database, the in-house database would need to include the 11 type strains of the valid species identified to date and the 7 new species detected in the study.
- There is no information guaranteeing the genetic diversity of the clinical CF strains studied (please specify the total number of patients and add information in supplementary table S1: patient number, ST if available). Without precision, it cannot be excluded that the same clone was analysed and this concerns many isolates (11 isolates of *R. insidiosa* from Turkey, 37 *R. mannitolilytica* from the United Kingdom, 24 from the United States) .

Minor comments

- For the database evaluation, sometimes the same isolate was used for the creation of the MSP and for identification, which necessarily contributes to a better performance of the database.
- Almost all the isolates used to build the database (creation of the MSPs) have been submitted to WGS, which is an asset for ensuring that the strains have been correctly identified. It would be better if possible to complete the WGS for the few isolates that are missing.
- the aim of the study is to improve diagnosis in human infections. Specify the number of clinical strains (distinguishing from hospital environmental strains) throughout the text
- Line 49 -51 : please cite the 11 species described to date.
- line 53 : "under 10%" please be more specific, in the references cited, the prevalence is under 5%.
- lines 59-60, (or in Methods) it would be useful to list the species included in the version of Bruker IVD db MSP-11758 used in the study (even if it is in the discussion part)
- line 67 : please precise the number of clinical (CF, non CF) and environmental strains, and the number of patients involved
- lines 70-71, please specify the number of novel species detected in clinical samples
- line 77 : please specify the method of identification of the isolates selected for the study.
- line 89 : the in-house RUO MSP-5331 database : was it built for this study or is it an existing library that was updated with the 32 MSP of this study ?
- line 91-92 : Why using these criteria ? These are not the current standard Bruker interpretative criteria. A score > 2.3 is not necessary for species identification, a score >2 with consistency category A can be accepted for species-level identification. It may be more appropriate to report the identification performances of the Bruker IVD database using the Bruker criteria.
- lines 105-106 : "Final MALDI-TOF MS identification was based on Bruker identification scores in combination with clustering results." Did the authors mean final identification and not final MS identification ? (suppl table S2 , column B or columns E and F ?). Please clarify.

- line 171 : please precise version or year of EUCAST clinical breakpoints.
- lines 199-201 : In total, If I am not mistaken, all *Ralstonia* species (11 described to date + 7 novel species) are not represented in Bruker IVD + in house databases . Please list the species included in the updated db .
- lines 193-214 : Supplementary Table S2 is very important for understanding and could be included in the body of the article - it is not clear how MS clustering was used to identify isolates, perhaps the authors should specify in methods that isolates similarly clustered by both techniques were considered to belong to the same species ? However, in the other cases, how did the authors conclude? It seems that clustering with SpeDe was not sufficient to distinguish species: within the S1 cluster, several species could be identified. So how did the authors conclude that isolate LMG 1811 belonged to *R. mannitolilytica* ?
- line 246 : please specify the number of clinical isolates included, and the number of CF isolates out of the 30. Figure 3 : is it possible to include this information ? (clinical /environmental)
- lines 246-250 the 2 recently described species are missing in the text (*chuxiongensis* and *wenshanensis*)
- line 259 : please precise (or in Methods) which were the 19 isolates /species analysed ? (this information is only available in supplementary Figure)
- line 264 : As these species are rarely detected in diagnostic laboratories, it is useful to specify that all grow at 37{degree sign}C and to specify the growth time required on conventional media (Mac Conkey, Drigalski) (18, 24, 48 h?).
- .-lines 297-298 and abstract : It is important to specify the number of novel species detected in human samples (and to distinguish from hospital environment), and in particular in patients with cystic fibrosis..
- lines 301-313 : please discuss the limitations of the in-house database associated with Bruker IVD db. The misidentifications detected can be due to Bruker incorrect species identification and to the lack of inclusion of several species.
- line 363 : CeoAB-OpcM efflux provides resistance to FQ and AG in *Burkholderia*. Please precise "in *Burkholderia cenocepacia*". How high is the degree of similarity between the efflux pump detected in the genomes and CeoAB-OpcM from *B. cenocepacia* ?
- line 367 : please precise if the identification of the isolate 16028 is accurate, or discuss performing WGS
- line 374 : please specify the betalactams antibiotics concerned in the text.

In the work titled "*Improved matrix-assisted laser desorption/ionization time-of-flight mass spectrometry-based identification and antimicrobial resistance patterns of Ralstonia recovered from human infections*" by Stephanie Steyaerta et al., presents an interesting analysis of a collection of 161 clinical and environmental *Ralstonia* isolates. The authors subjected this population to a MALDI-TOF dereplication and identification using the commercial Bruker database (IVD V12.0, MSP-11758). The identification scores allowed the confident identification of only 62 isolates (39%) down to the species level.

Based on this result they selected 32 representative isolates for which no confident MALDI-TOF MS species level identification was achieved, to accomplish a Whole-genome sequence analysis. The study revealed the presence of seven novel *Ralstonia* species from clinical and environmental sources. Using these novel sequences and the sequences of other 6 known species they developed an improved *in-house* MALDI TOF database (RUO, MSP-5331). A reanalysis of the original *Ralstonia* population, using this updated MALDI-TOF database increased the percentage of isolates with confident species level identification up to 77%.

The authors analyzed the antimicrobial resistance genes in 43 genomes comprising the novel genomes reported in this investigation and the genomes of 11 established *Ralstonia* species demonstrating a broadly distributed beta-lactam resistance determinants within the genus *Ralstonia*. In addition, they determined the antimicrobial susceptibility of selected human clinical and environmental *Ralstonia* isolates against 17 antimicrobial agents. Most of the novel *Ralstonia* species were multi-resistant, but most of the population is still susceptible to trimethoprim /sulfamethoxazole, ciprofloxacin and tigecycline.

Ralstonia species, can cause significant disease in certain at-risk patient groups, including those with cystic fibrosis. Challenges in accurate clinical laboratory identification arise due to frequent misidentification with closely related bacteria, the dynamic taxonomy within this genus, their notable multidrug-resistant profiles, and species-specific variations in antimicrobial resistance. Consequently, conducting systematic research aimed at enhancing identification methods and comprehensively analyzing antimicrobial resistance becomes imperative. Hence, the findings presented here represent a significant leap forward in our understanding of this genus."

Ralstonia species, can cause significant disease in certain at-risk patient groups, including those with cystic fibrosis. Challenges in accurate clinical laboratory identification arise due to frequent misidentification with closely related bacteria, the dynamic taxonomy within this genus, their notable multidrug-resistant profiles, and species-specific variations in antimicrobial resistance. Consequently, conducting systematic research aimed at enhancing identification methods and comprehensively analyzing antimicrobial resistance

becomes imperative. Hence, the findings presented here substantially advance our understanding of the *Ralstonia* genus.

The work is very well organized, the methodology is clear and robust.

Minor concerns:

Title: as MALDI-TOF-MS has also been applied in the detection of drug-resistance mechanisms and particularly the detection of resistance against β -Lactam antibiotics, I would propose:

Improved matrix-assisted laser desorption/ionization time-of-flight mass spectrometry-based identification and analysis of antimicrobial resistance patterns of *Ralstonia* recovered from human infections

However, from my perspective I believe this title does not fully capture the main contributions of this manuscript, which I believe involve, the improvement of the *Ralstonia* MALDI-TOF MS-based identification primarily through the incorporation of seven novel *Ralstonia* species into the commercial database, and the analysis of antimicrobial resistance patterns of *Ralstonia* recovered from human infections. Wouldn't it be important to mention in the title that this investigation describes seven novel *Ralstonia* species? The improvement in MALDI-TOF-MS identification in this work does not lie on the MALDI-TOF MS methodology itself but rather on the development of an updated database.

Results section:

Lines 199 to 222: Given that the number of isolates selected for WGS identification was 32 (indicated in Supplementary table S2) it coincides with the number of isolates used to update the *in-house* MALDI-TOF database (presented in Supplementary table S3), it might be advisable to alert the reader in line 199: "The *in-house* MALDI-TOF MS database was subsequently updated to include MSPs of (i) 19 isolates of 6 established *Ralstonia* species, including *R. insidiosa*, *R. mannitolilytica*, *R. pickettii*, *R. pseudosolanacearum*, *R. solanacearum*, and 3 subspecies of *R. syzygii*, (ii) 2 isolates of recently described species that were missing in the commercial Bruker database (IVD V12.0, MSP-11758), and (iii) 11 isolates of the 7 novel species detected in the present study (Supplementary table S3).

Discussion section

Line 314. The authors could include in the discussion whether the identification at the genus level or the misidentification of certain isolates could be attributed to the low availability of certain isolates mainly among the newly incorporated species (*e.g.* for *R. flaminis* and *R. condita* only one representative isolate of each species was available to be incorporated into the database).

Spectrum04021-23

Original title: Improved matrix-assisted laser desorption/ionization time-of-flight mass spectrometry-based identification and antimicrobial resistance patterns of *Ralstonia* recovered from human infections

Revised title: Novel *Ralstonia* species from human infections: improved matrix-assisted laser desorption/ionization time-of-flight mass spectrometry-based identification and analysis of antimicrobial resistance patterns

Response to Reviews

Reviewer #1

Several years ago Bruker changed the score classification and this species and genus level identification classification is no longer in place (lines 91-94) and the manufacturer classifies them as: (2.00 - 3.00) High Confidence Identification, (1.70 - 1.99) Low Confidence Identification and (0.00 - 1.69) No Organism Identification Possible. How would the performance vary using the current definitions for both the commercial and the in-house library.

The score classification system mentioned by the referee is used in the MBT Compass Client software and not in MBT Compass Explorer. We had written in L89 of the original text (L98 of the revised manuscript with marked changes) that we used MBT Compass Explorer software to examine all MALDI-TOF MS identification results; and hence applied the corresponding score classification system. We explicitly added this information to the manuscript when listing the score classification system to clarify and emphasize this further (L100-101 of the revised manuscript with marked changes). By changing the score classification system to the one mentioned above and thus by using a threshold of 2.0 instead of 2.3 for high confidence identification, a considerable larger number of isolates would have been identified at the genus level only, because more isolates would have scores > 2.0 with more than one species, thus necessitating a dramatic increase in further genome-based identifications. While once could intuitively expect otherwise, applying a more strict threshold allowed to more efficient species level identifications. We also had addressed explicitly the use of these thresholds in the discussion of the original manuscript (now L321-335 of the revised manuscript with marked changes) as it was one of the aims of the present study was to improve the identification of *Ralstonia* via MALDI-TOF MS, not only by expanding our reference database, but also by examining the usefulness of the identification score thresholds.

*Why were the *Pseudomonas* spp. breakpoints for Aminoglycosides used instead of the PK/PD ones?*

For aminoglycosides, we used the PK/PD breakpoints for *Pseudomonas* because our ranges of testing were 4-32 µg/ml for amikacin (PK/PD breakpoint 1/1) and 1-16 µg/ml for tobramycin (PK/PD breakpoint 0.5/0.5) and our testing range was higher than the PK/PD breakpoints. We accept it can be criticized to assume that *Ralstonia* bacteria have the same resistance for aminoglycosides as *Pseudomonas* bacteria and therefore modified our analysis. For amikacin and tobramycin the MIC results were now interpreted according to the general PK/PD breakpoints and Figure 3 was adapted accordingly. For MIC values of <4 µg/ml for amikacin and <1 µg/ml for tobramycin we thus could not conclude whether the strains were S, R or I (Figure 3 was adapted accordingly). Nevertheless, the overall conclusion that *Ralstonia* bacteria were generally highly resistant to amikacin and tobramycin did not change.

*The use of species (as in *Ralstonia* species) and *Ralstonia* sp. instead of *Ralstonia* spp. should be reviewed through the whole manuscript (as in line 285 for example).*

*The same situation occurs with *Pseudomonas* spp. and *Acinetobacter* spp. (lines 175-176 and 499-500).*

We are unsure what the reviewer points to. We did not use the abbreviation "spp." in the

manuscript. We used the “sp.” designation, such as *Ralstonia* sp., to denote a genus level identification (e.g. L204 of the revised manuscript with marked changes). We used “*Ralstonia* species” to refer to one or more ‘species’ within the genus *Ralstonia* (e.g. L72 of the revised manuscript with marked changes).

Reviewer #2

In the work titled "Improved matrix-assisted laser desorption/ionization time-of-flight mass spectrometry-based identification and antimicrobial resistance patterns of Ralstonia recovered from human infections" by Stephanie Steyaert et al., presents an interesting analysis of a collection of 161 clinical and environmental Ralstonia isolates. The authors subjected this population to a MALDI-TOF dereplication and identification using the commercial Bruker database (IVD V12.0, MSP-11758). The identification scores allowed the confident identification of only 62 isolates (39%) down to the species level.

Based on this result they selected 32 representative isolates for which no confident MALDI-TOF MS species level identification was achieved, to accomplish a Whole-genome sequence analysis. The study revealed the presence of seven novel Ralstonia species from clinical and environmental sources. Using these novel sequences and the sequences of other 6 known species they developed an improved in-house MALDI TOF database (RUO, MSP-5331). A reanalysis of the original Ralstonia population, using this updated MALDI-TOF database increased the percentage of isolates with confident species level identification up to 77%.

The authors analyzed the antimicrobial resistance genes in 43 genomes comprising the novel genomes reported in this investigation and the genomes of 11 established Ralstonia species demonstrating a broadly distributed beta-lactam resistance determinants within the genus Ralstonia. In addition, they determined the antimicrobial susceptibility of selected human clinical and environmental Ralstonia isolates against 17 antimicrobial agents. Most of the novel Ralstonia species were multi-resistant, but most of the population is still susceptible to trimethoprim /sulfamethoxazole, ciprofloxacin and tigecycline.

Ralstonia species, can cause significant disease in certain at-risk patient groups, including those with cystic fibrosis. Challenges in accurate clinical laboratory identification arise due to frequent misidentification with closely related bacteria, the dynamic taxonomy within this genus, their notable multidrug-resistant profiles, and species-specific variations in antimicrobial resistance. Consequently, conducting systematic research aimed at enhancing identification methods and comprehensively analyzing antimicrobial resistance becomes imperative. Hence, the findings presented here represent a significant leap forward in our understanding of this genus.

Minor concerns

Title: as MALDI-TOF-MS has also been applied in the detection of drug-resistance mechanisms and particularly the detection of resistance against β -Lactam antibiotics, I would propose:

Improved matrix-assisted laser desorption/ionization time-of-flight mass spectrometry-based identification and analysis of antimicrobial resistance patterns of Ralstonia recovered from human infections

However, from my perspective I believe this title does not fully capture the main contributions of this manuscript, which I believe involve, the improvement of the Ralstonia MALDI-TOF MS-based identification primarily through the incorporation of seven novel Ralstonia species into the commercial database, and the analysis of antimicrobial resistance patterns of Ralstonia recovered from human infections. Wouldn't it be important to mention in the title that this investigation describes seven novel Ralstonia species? The improvement in MALDI-TOF-MS identification in this work does not lie on the MALDI-TOF MS methodology itself but rather on the development of an updated database.

While we agree with the referee it proved difficult to combine all relevant information in a concise and comprehensive title. After numerous versions we propose to modify the title to:

“Novel *Ralstonia* species from human infections: improved matrix-assisted laser desorption/ionization time-of-flight mass spectrometry-based identification and analysis of antimicrobial resistance patterns”

We are nevertheless open to suggestions.

Results section

Lines 199 to 222: Given that the number of isolates selected for WGS identification was 32 (indicated in Supplementary table S2) it coincides with the number of isolates used to update the in-house MALDI-TOF database (presented in Supplementary table S3), it might be advisable to alert the reader in line 199: "The in-house MALDI-TOF MS database was subsequently updated to include MSPs of (i) 19 isolates of 6 established Ralstonia species, including R. insidiosa, R. mannitolilytica, R. pickettii, R. pseudosolanacearum, R. solanacearum, and 3 subspecies of R. syzygii, (ii) 2 isolates of recently described species that were missing in the commercial Bruker database (IVD V12.0, MSP-11758), and (iii) 11 isolates of the 7 novel species detected in the present study (Supplementary table S3).

We agree and revised the text accordingly (L211-216 of the revised manuscript with marked changes).

Discussion section

Line 314. The authors could include in the discussion whether the identification at the genus level or the misidentification of certain isolates could be attributed to the low availability of certain isolates mainly among the newly incorporated species (e.g. for R. flaminis and R. condita only one representative isolate of each species was available to be incorporated into the database).

We agree and revised the text accordingly (L333-335 of the revised manuscript with marked changes).

Reviewer #3

This is a very interesting article on the identification of Ralstonia species. Ralstonia spp are pathogens for humans or plants, depending on the species concerned, and are considered to be emerging in cystic fibrosis patients. Epidemiological data is lacking because these pathogens are rarely isolated from human samples and identification techniques are not reliable for the species. Using a collection of 161 isolates, the authors highlight the limitations of the commercially available MALDI-TOF MS Bruker database for accurate species identification and suggest that implementation of the existing database would improve results. In addition, the authors describe 7 novel species within the genus, 4 of which were detected from human samples. Finally, the authors report antimicrobial resistance profiles of 30 isolates belonging to 12 species, providing new data.

Major comments

-The reference method used to assess the performance of MALDI-TOF MS to identify isolates is not provided (apart from reference strains of established species, or WGS for 32 isolates). For the other isolates, which identification method was used?

For historical reference isolates, we referred to previous taxonomic studies as shown in

Supplementary Table S1. We added the required references for all other isolates to the methods section (L88-89 of the revised manuscript with marked changes).

-The authors suggest that the isolate identifications used in the Bruker IVD database may be erroneous (this has already been described for other bacterial genera). However, they still used a combination of the Bruker IVD database and their own database to identify isolates. This could therefore lead to erroneous identifications. To build an accurate identification database, the in-house database would need to include the 11 type strains of the valid species identified to date and the 7 new species detected in the study.

The first main aim of our study was to improve identification of *Ralstonia* isolates from human samples and the hospital environment, as this is where our main interest is situated. We agree that a complete database sooner or later has to include all environmental species also. The species that are missing in our combined databases are *R. soli*, *R. mojiangensis* and *R. nicotianae*, but our genomic analyses demonstrated that these were not present in our collections. We also agree that having a single reference library with all spectra would be more efficient; yet are confident that it would not be more effective.

*-There is no information guaranteeing the genetic diversity of the clinical CF strains studied (please specify the total number of patients and add information in supplementary table S1: patient number, ST if available). Without precision, it cannot be excluded that the same clone was analysed and this concerns many isolates (11 isolates of *R. insidiosa* from Turkey, 37 *R. mannitolilytica* from the United Kingdom, 24 from the United States) .*

We do not understand why this issue is raised. Our study was not meant or designed to report or evaluate the genetic diversity of CF isolates, nor to cover the diversity seen in one CF center, in one country or globally. While per CF patient only a single isolate was included (this information was added to the revised text on L88-89), we have no detailed patient information, nor a view on the number of patients with or without *Ralstonia*, per center or per country. We did not discuss prevalence of different *Ralstonia* species per center, per country or globally; this is not what this paper is about. It is indeed possible that different isolates represent the same strain/clone, as indeed the multiple *R. insidiosa* isolates from Turkish patients, but this is why we consistently used the term 'isolate' over 'strain' in the text.

Minor comments

-For the database evaluation, sometimes the same isolate was used for the creation of the MSP and for identification, which necessarily contributes to a better performance of the database.

This is correct.

-Almost all the isolates used to build the database (creation of the MSPs) have been submitted to WGS, which is an asset for ensuring that the strains have been correctly identified. It would be better if possible to complete the WGS for the few isolates that are missing.

Six isolates (including three that represent plant pathogenic species that have never been reported in human samples) were used for MSP creation while there was no genome sequence available: *R. mannitolilytica* R-76692, *R. mannitolilytica* R-76714, *R. pickettii* LMG 7003, *R. pseudosolanacearum* LMG 31754, *R. pseudosolanacearum* LMG 31756 and *R. solanacearum* LMG 17143. Four of these isolates represent taxonomic reference strains and for all 6 isolates, the MALDI-TOF MS identification results were unambiguous (both in terms of Bruker identification score and clustering). We are therefore confident that a genome sequence analysis was not necessary.

-the aim of the study is to improve diagnosis in human infections. Specify the number of clinical strains (distinguishing from hospital environmental strains) throughout the text

This was clarified in the manuscript (L29-32 (abstract) and L73-75 (introduction) of the revised manuscript with marked changes).

-Line 49 -51 : please cite the 11 species described to date.

This information was added to the manuscript (L54-57 of the revised manuscript with marked changes).

-line 53 : "under 10%" please be more specific, in the references cited, the prevalence is under 5%.

We changed the number to "under the 5%" (L59 of the revised manuscript with marked changes).

-lines 59-60, (or in Methods) it would be useful to list the species included in the version of Bruker IVD db MSP-11758 used in the study (even if it is in the discussion part)

This information was added to the Results section (L201-202 of the revised manuscript with marked changes) and Supplementary table S3.

- line 67 : please precise the number of clinical (CF, non CF) and environmental strains, and the number of patients involved

This information was added to the manuscript (L29-32 (abstract) and L73-75 (introduction) of the revised manuscript with marked changes).

-lines 70-71, please specify the number of novel species detected in clinical samples

This information was added to the manuscript (L37-38 (abstract), L78-79 (introduction) and L316-318 (discussion) of the revised manuscript with marked changes).

- line 77 : please specify the method of identification of the isolates selected for the study.

For historical reference isolates, we refer to taxonomic studies in which these bacteria were described and named (Supplementary table S1); for other isolates we provided additional references in the text (L88-89 of the revised manuscript with marked changes).

-line 89 : the in-house RUO MSP-5331 database : was it built for this study or is it an existing library that was updated with the 32 MSP of this study ?

Our in-house MALDI-TOF MS reference database comprising > 5,000 MSPs was an existing reference database that was updated for the present study with an extra 32 MSPs of *Ralstonia* isolates.

- line 91-92 : Why using these criteria ? These are not the current standard Bruker interpretative criteria. A score > 2.3 is not necessary for species identification, a score >2 with consistency category A can be accepted for species-level identification. It may be more appropriate to report the identification performances of the Bruker IVD database using the Bruker criteria.

Bruker nowadays indeed promotes another score classification system but we disagree with the statement of the reviewer that a score >2 with consistency category A can be accepted for species-level identification. The Bruker MBT Compass manual states the following:
1.1.2 Calculating a log(score) value Data acquisition is controlled using MBT Compass software.

[...] The higher the log(score), the higher the degree of similarity between the pattern for the unknown peak list and the peak list for the database entry in the reference library. A log (score) greater than or equal to 2.000 is considered an acceptable probability for sample identification at the species level. If the log(score) is less than 2.000 after initial analysis, samples can be processed using an alternative sample preparation procedure and the analysis repeated. For more information on alternative sample preparation procedures, see Appendix C. The log (score) ranges defined in MBT Compass reflect the probability of organism identification. Results should be reviewed by a trained microbiologist and final organism identification should be based on all relevant information available. This information includes but is not limited to: Gram staining, colony morphology, growth characteristics, and sample matrix.

Comment: Additional information that may be useful to achieve a greater level of confidence in an identification. The same information is provided in the Matching Hints section of the results report. Consistency: This additional information may be useful to achieve a greater overall level of confidence in the identification. Exemplary combinations of results that generate each consistency category are listed in Appendix D.

We do agree that a threshold of 2.0 is commonly used and promoted today by Bruker but it is depicted as such in the MBT Compass Client software only and not in MBT Compass Explorer. As stated in L89 (L98 of the revised manuscript with marked changes), we used MBT Compass Explorer software to examine all MALDI-TOF MS identification results and we described the score classification system the way it was applied and depicted in the software used. We added this information to the manuscript when listing the score classification system to clarify this further (L100-101 of the revised manuscript with marked changes).

By changing the score classification system and thus by using a threshold of 2.0 instead of 2.3 for high confidence identification, a considerable larger number of isolates would have been identified at the genus level only, because more isolates would have scores > 2.0 **with more than one species**, thus necessitating a dramatic increase in further genome-based identifications. While one could intuitively expect otherwise, applying a more strict threshold allowed to more efficient species level identifications. We also had addressed explicitly the use of these thresholds level in the discussion of the original manuscript (now L321-335 of the revised manuscript with marked changes) as it was one of the aims of the present study was to improve the identification of *Ralstonia* via MALDI-TOF MS, not only by expanding our reference database, but also by examining the usefulness of the Bruker identification score thresholds.

-lines 105-106 : "Final MALDI-TOF MS identification was based on Bruker identification scores in combination with clustering results." Did the authors mean final identification and not final MS identification ? (suppl table S2 , column B or columns E and F ?). Please clarify.

Columns E and F show the MALDI-TOF MS identification based on analysis of the Bruker ID score in reference to only the Bruker IVD database (column E) or the Bruker IVD and our in-house reference databases together (column F). Column B represents the final MALDI-TOF MS identification based on the analysis of the Bruker ID scores and the clustering analyses.

We added this information to the legend of Supplementary table S2 and changed its title accordingly to clarify this.

-line 171 : please precise version or year of EUCAST clinical breakpoints.

This information was added to the manuscript (L181 of the revised manuscript with marked changes).

-lines 199-201 : In total, If I am not mistaken, all Ralstonia species (11 described to date + 7 novel species) are not represented in Bruker IVD + in house databases . Please list the species included

in the updated db .

We added the exact list of MSPs that were present in Bruker IVD V12.0 (MSP-11758) to Supplementary table S3 and this table now comprises a complete overview of MSPs in the Bruker IVD and our in-house database.

As a result of the update, all species were present except the recently described *R. soli*, *R. mojiangensis* and *R. nicotianae*. We assume that Bruker will add these novel validly named species to its reference databases.

-lines 193-214 : Supplementary Table S2 is very important for understanding and could be included in the body of the article

We agree this table is important for the manuscript. However, it is easier to sort, filter and search when giving it as an excel table in supplementary material to the reader. If the editor is, however, also convinced that this table should be transferred to the main manuscript, we are willing to update this before publication.

*- it is not clear how MS clustering was used to identify isolates, perhaps the authors should specify in methods that isolates similarly clustered by both techniques were considered to belong to the same species ? However, in the other cases, how did the authors conclude? It seems that clustering with SpeDe was not sufficient to distinguish species: within the S1 cluster, several species could be identified. So how did the authors conclude that isolate LMG 1811 belonged to *R. mannitolilytica* ?*

The MALDI-TOF MS analyses not only included identification based on the Bruker identification scores but also dereplication, i.e. clustering based on the similarity (BioNumerics) or unicity (SPeDE) of MALDI-TOF mass spectra. Final MALDI-TOF MS identification was thus based on Bruker identification scores in combination with the clustering of MALDI-TOF mass spectra. In case the two replicate spectra from an isolate belonged to a cluster that contained spectra that were identified with high confidence, that isolate was identified as the same species.

Finally, we appreciated the question regarding SPeDE cluster S1 (Supplementary table S2) as it revealed a mistake in our earlier analyses which needed correction. This mistake was provoked by not completely re-running the SPeDE analysis on the final dataset for the present study. We therefore decided to re-ran SPeDE on the complete dataset for the present revision (i.e. including duplicate mass spectra for 161 isolates) and found a considerable higher number of SPeDE clusters. In addition to SPeDE cluster S1, also SPeDE cluster S2 and S5 were now split into smaller clusters (all cluster numbers were accordingly revised, i.e. in L205 of the revised manuscript with marked changes and Supplementary table S2), and these novel SPeDE groupings corresponded far better to the clusters delineated through the spectrum similarity-based analysis obtained with the BioNumerics software. The latter BioNumerics clustering had already been used (in the original version of the text) as the starting point for selecting isolates for whole-genome sequencing, so the discovery of a larger number of 'smaller groupings' through the SPeDE analysis had no further impact on the identification of isolates in subsequent analyses.

-line 246 : please specify the number of clinical isolates included, and the number of CF isolates out of the 30. Figure 3 : is it possible to include this information ? (clinical /environmental)

We added a column to Supplementary table S1 to give an overview of the isolates used for MIC analyses and to retrieve the data about isolation source. As Figure 3 already contains much information, we prefer not to add the isolation source information to this figure as well. We fear the figure would become too 'crowded' which would render the most important data less readable.

*-lines 246-250 the 2 recently described species are missing in the text (*chuxiongensis* and *wenshanensis*)*

This was corrected (L262-263 of the revised manuscript with marked changes).

-line 259 : please precise (or in Methods) which were the 19 isolates /species analysed ? (this information is only available in supplementary Figure)

This information was added to the manuscript (L275-278 of the revised manuscript with marked changes). We also added a column to Supplementary table S1 to give an overview of the isolates used for phenotyping.

-line 264 : As these species are rarely detected in diagnostic laboratories, it is useful to specify that all grow at 37{degree sign}C and to specify the growth time required on conventional media (Mac Conkey, Drigalski) (18, 24, 48 h?).

This information was added to the manuscript (L283 of the revised manuscript with marked changes).

.-lines 297-298 and abstract : It is important to specify the number of novel species detected in human samples (and to distinguish from hospital environment), and in particular in patients with cystic fibrosis.

This information was added to the manuscript (L37-38 (abstract), L78-79 (introduction) and L316-318 (discussion) of the revised manuscript with marked changes).

-lines 301-313 : please discuss the limitations of the in-house database associated with Bruker IVD db. The misidentifications detected can be due to Bruker incorrect species identification and to the lack of inclusion of several species.

We think this comment corresponds with the last comment of reviewer 2; and revised the text accordingly (L333-335 of the revised manuscript with marked changes).

-line 363 : CeoAB-OpcM efflux provides resistance to FQ and AG in Burkholderia. Please precise "in Burkholderia cenocepacia". How high is the degree of similarity between the efflux pump detected in the genomes and CeoAB-OpcM from B. cenocepacia ?

This information was added to the manuscript (L386 of the revised manuscript with marked changes). The degree of similarity towards the reference genes in CARD is depicted in Figure 2.

-line 367 : please precise if the identification of the isolate 16028 is accurate, or discuss performing WGS

We revised the text accordingly (L391-392 of the revised manuscript with marked changes).

-line 374 : please specify the betalactams antibiotics concerned in the text.

This information was added to the manuscript (L397 of the revised manuscript with marked changes).

Re: Spectrum04021-23R1 (Novel *Ralstonia* species from human infections: improved matrix-assisted laser desorption/ionization time-of-flight mass spectrometry-based identification and analysis of antimicrobial resistance patterns)

Dear Prof. Peter VANDAMME:

I am pleased to inform you that your manuscript is nearly accepted for publication. However, there are a few minor points that need to be addressed. Once these are completed, please return your submission so that I can move your paper forward to acceptance.

Sincerely,
Silvia Cardona
Editor
Microbiology Spectrum

Reviewer #1 (Comments for the Author):

I still think that the use of spp. and sp. thorough the manuscript is sometimes mistaken as in line 184 and 186 and 516-517 in the marked manuscript as the EUCAST Breakpoints are for *Pseudomonas* spp. and *Acinetobacter* spp. not for one sp. in particular. However, I don't think this is a problem that diminishes the quality of the manuscript whatsoever.

Reviewer #2 (Comments for the Author):

Minor comments

-It is preferable to avoid using the term 'cystic fibrosis patients' or 'CF patients'. Instead, I would recommend 'people with cystic fibrosis', 'patients with cystic fibrosis', or 'individuals with cystic fibrosis', as these phrases emphasize the humanity of the person rather than defining them by their medical condition.

-There are several paragraphs, particularly the ones added in this new version, where the number of the different species of *Ralstonia* is indicated in the same sentence, in both numerical digits and written-out words. As an example, in the first sentence of the Abstract it is said "...collection of **161** *Ralstonia* isolates, including 90 isolates from cystic fibrosis patients, 27 isolates from other human clinical samples, eight isolates from the hospital environment, seven isolates from industrial samples and 19 environmental isolates..." The same in Lines 80 to 82.

Reviewer #3 (Comments for the Author):

The authors have improved the manuscript and answered all the questions raised.

Only one point should be clarified in the manuscript.

The question of the genetic diversity of the isolates was raised because the aim of the study was to improve the identification at species level by MALDI-TOF MS. The performance of the database depends on the diversity of the strains used within a same species. This is therefore important information to report in my opinion.

If only one isolate per patient was included, this most likely represents different strains. However, if several isolates were used for the same patient, they are most likely the same strain.

This should be stated more clearly in the text : L88-89 "they represented different patients" does not necessarily mean a single isolate per patient. Please amend the text.

Spectrum04021-23

Revised title: Novel *Ralstonia* species from human infections: improved matrix-assisted laser desorption/ionization time-of-flight mass spectrometry-based identification and analysis of antimicrobial resistance patterns

Response to Reviews

Reviewer #1

I still think that the use of spp. and sp. thorough the manuscript is sometimes mistaken as in line 184 and 186 and 516-517 in the marked manuscript as the EUCAST Breakpoints are for Pseudomonas spp. and Acinetobacter spp. not for one sp. in particular. However, I don't think this is a problem that diminishes the quality of the manuscript whatsoever.

We modified this into *Pseudomonas* spp. and *Acinetobacter* spp. (L191-192 of the revised manuscript with marked changes and legend of Figure 3).

Reviewer #2

-It is preferable to avoid using the term 'cystic fibrosis patients' or 'CF patients'. Instead, I would recommend 'people with cystic fibrosis', 'patients with cystic fibrosis', or 'individuals with cystic fibrosis', as these phrases emphasize the humanity of the person rather than defining them by their medical condition.

We agree and revised the text into 'persons with cystic fibrosis' (L29, L80 and L378 of the revised manuscript with marked changes).

-There are several paragraphs, particularly the ones added in this new version, where the number of the different species of Ralstonia is indicated in the same sentence, in both numerical digits and written-out words. As an example, in the first sentence of the Abstract it is said "...collection of 161 Ralstonia isolates, including 90 isolates from cystic fibrosis patients, 27 isolates from other human clinical samples, eight isolates from the hospital environment, seven isolates from industrial samples and 19 environmental isolates..." The same in Lines 80 to 82.

We revised this and made sure that in sentences with an enumeration all numbers were written in the same format, thus either as numerical digits (L30 and L81 of the revised manuscript with marked changes) or in full (L242 and L320 of the revised manuscript with marked changes).

Reviewer #3

The authors have improved the manuscript and answered all the questions raised.

Only one point should be clarified in the manuscript.

The question of the genetic diversity of the isolates was raised because the aim of the study was to improve the identification at species level by MALDI-TOF MS. The performance of the database depends on the diversity of the strains used within a same species. This is therefore important information to report in my opinion.

If only one isolate per patient was included, this most likely represents different strains. However, if several isolates were used for the same patient, they are most likely the same strain.

This should be stated more clearly in the text : L88-89 "they represented different patients" does not necessarily mean a single isolate per patient. Please amend the text.

We corrected this in the manuscript (L94-95 of the revised manuscript with marked changes).

Re: Spectrum04021-23R2 (Novel *Ralstonia* species from human infections: improved matrix-assisted laser desorption/ionization time-of-flight mass spectrometry-based identification and analysis of antimicrobial resistance patterns)

Dear Dr. Charlotte Peeters:

Your manuscript has been accepted, and I am forwarding it to the ASM production staff for publication. Your paper will first be checked to make sure all elements meet the technical requirements. ASM staff will contact you if anything needs to be revised before copyediting and production can begin. Otherwise, you will be notified when your proofs are ready to be viewed.

Sincerely,
Silvia Cardona
Editor
Microbiology Spectrum